# Enhancing Competencies and Professional Upskilling of Mobile Healthcare Unit Personnel at the Hellenic National Public Health Organization

**DOI:** 10.3390/healthcare13141706

**Published:** 2025-07-15

**Authors:** Marios Spanakis, Maria Stamou, Sofia Boultadaki, Elias Liantis, Christos Lionis, Georgios Marinos, Anargiros Mariolis, Andreas M. Matthaiou, Constantinos Mihas, Varvara Mouchtouri, Evangelia Nena, Efstathios A. Skliros, Emmanouil Smyrnakis, Athina Tatsioni, Georgios Dellis, Christos Hadjichristodoulou, Emmanouil K. Symvoulakis

**Affiliations:** 1Department of Social Medicine, School of Medicine, University of Crete, 70013 Heraklion, Greece; lionis@galinos.med.uoc.gr; 2Center for Training and Lifelong Learning, University of Crete, 70013 Heraklion, Greece; 3Hellenic National Public Health Organization, 11527 Athens, Greece; m.stamou@eody.gov.gr (M.S.); covid.reg33@eody.gov.gr (S.B.); i.liantis@eody.gov.gr (E.L.); g.dellis@eody.gov.gr (G.D.); or xhatzi@uth.gr (C.H.); 4Department of Hygiene, Epidemiology and Medical Statistics, Medical School, National and Kapodistrian University of Athens, 11527 Athens, Greece; gmarino@med.uoa.gr; 5Department of Primary Care, Health Center of Areopolis, 23062 Areopolis, Greece; amariolis@gmail.com; 65th Department of Respiratory Medicine, Sotiria Thoracic Diseases General Hospital of Athens, 11527 Athens, Greece; matthaiou.andreas@gmail.com; 7General Hospital-Health Centre of Kymi, 34003 Kymi, Greece; gas521@yahoo.co.uk; 8Department of Hygiene and Epidemiology, School of Medicine, University of Thessaly, 41222 Larissa, Greece; mouchtourib@med.uth.gr; 9Laboratory of Social Medicine, Medical School, Democritus University of Thrace, 68131 Alexandroupolis, Greece; enena@med.duth.gr; 10Nemea Health Center, 20100 Corinthia, Greece; stathis.skliros@gmail.com; 11Laboratory of Primary Health Care, General Practice and Health Services Research, School of Medicine, Aristotle University of Thessaloniki, 62121 Thessaloniki, Greece; smyrnak@auth.gr; 12Department of Research for General Medicine and Primary Health Care, School of Health Sciences, Faculty of Medicine, University of Ioannina, 45500 Ioannina, Greece; atatsion@uoi.gr; 13Laboratory of Hygiene and Epidemiology, Faculty of Medicine, University of Thessaly, 62100 Larissa, Greece

**Keywords:** mobile healthcare units, community health, public health training, primary healthcare, public health, online learning, health education, knowledge acquisition, continuing medical education, competency-based training, upskilling

## Abstract

**Background/Objectives:** Mobile healthcare units (MHUs) comprise flexible, ambulatory healthcare teams that deliver community care services, particularly in underserved or remote areas. In Greece, MHUs were pivotal in epidemiological surveillance during the COVID-19 pandemic and are now evolving into a sustainable and integrated service for much-needed community-based healthcare. To support this expanded role, targeted, competency-based training is essential; however, this can pose challenges, especially in coordinating synchronous learning across geographically dispersed teams and in ensuring engagement using an online format. **Methods:** A nationwide, online training program was developed to improve the knowledge of the personnel members of the Hellenic National Public Health Organization’s MHUs. This program was structured focusing on four core themes: (i) prevention–health promotion; (ii) provision of care; (iii) social welfare and solidarity initiatives; and (iv) digital health skill enhancement. The program was implemented by the University of Crete’s Center for Training and Lifelong Learning from 16 January to 24 February 2025. A multidisciplinary team of 64 experts delivered 250 h of live and on-demand educational content, including health screenings, vaccination protocols, biomarker monitoring, chronic disease management, treatment adherence, organ donation awareness, counseling on social violence, and eHealth applications. Knowledge acquisition was assessed through a pre- and post-training multiple-choice test related to the core themes. Trainees’ and trainers’ qualitative feedback was evaluated using a 0–10 numerical rating scale (Likert-type). **Results:** A total of 873 MHU members participated in the study, including both healthcare professionals and administrative staff. The attendance rate was consistently above 90% on a daily basis. The average assessment score increased from 52.8% (pre-training) to 69.8% (post-training), indicating 17% knowledge acquisition. The paired *t*-test analysis demonstrated that this improvement was statistically significant (*t* = −8.52, *p* < 0.001), confirming the program’s effectiveness in enhancing knowledge. As part of the evaluation of qualitative feedback, the program was positively evaluated, with 75–80% of trainees rating key components such as content, structure, and trainer effectiveness as “Very Good” or “Excellent.” In addition, using a 0–10 scale, trainers rated the program relative to organization (9.4/10), content (8.8), and trainee engagement (8.9), confirming the program’s strength and scalability in primary care education. **Conclusions:** This initiative highlights the effectiveness of a structured, online training program in enhancing MHU knowledge, ensuring standardized, high-quality education that supports current primary healthcare needs. Future studies evaluating whether the increase in knowledge acquisition may also result in an improvement in the personnel’s competencies, and clinical practice will further contribute to assessing whether additional training programs may be helpful.

## 1. Introduction

Primary healthcare (PHC) plays a fundamental role in strengthening health systems by delivering equitable, accessible, and person-centered services within communities [1,2]. Adhering to the principles of integration, intersectoral collaboration, and community engagement, PHC not only addresses individual health needs, but also broader social, economic, and environmental determinants of health [3,4]. Complementing PHC, public health (PH) focuses on population-wide health promotion, disease prevention, and evidence-based policymaking [2,5]. Recent global health challenges, such as the COVID-19 pandemic, have highlighted the urgent need to shift from reactive, hospital-centered care toward proactive, community-based approaches [6,7]. Notably, evidence shows that over 90% of COVID-19 cases were managed outside hospital settings, underscoring the importance of strong primary and community healthcare services [8]. This paradigm shift calls for a rebalancing of health systems from hospital-driven structures to integrated, community-responsive care models that are better suited to emerging local health needs [9,10,11]. According to the World Health Organization (WHO), upcoming health systems must be designed around people rather than institutions, emphasizing community involvement and upstream prevention to build resilient and equitable healthcare infrastructure [6,12].

Mobile healthcare units (MHUs) represent a key element in realizing this vision. As agile extensions of the health system, MHUs are uniquely positioned to act at the intersection of real-life needs and healthcare delivery, improving population health management and reducing costs [13,14]. Although MHUs gained prominence during the COVID-19 pandemic period, their utility has long been recognized, especially in disaster settings and conflict zones [15,16,17]. Furthermore, the WHO introduced the term emergency medical teams (EMTs), which can be deployed to aid populations affected by disasters [18,19]. In addition, the WHO has emphasized the role of mobile clinics in providing timely care to displaced and underserved populations [15,20,21]. On the other hand, the deployment of mobile units during the pandemic demonstrated their effectiveness in enhancing epidemiological surveillance and local response [16,22,23].

Today, MHUs—or mobile clinics—are recognized as flexible, community-based healthcare delivery teams that offer a range of services, including chronic disease management, preventive screening, health education, and social care [24]. In the United States, MHUs have proven effective in managing conditions such as asthma, diabetes, and hypertension, while reducing emergency room visits and improving care access [24,25,26,27,28]. European experiences similarly show that MHUs attract hard-to-reach groups and contribute to the early detection of non-communicable diseases [29]. MHUs can contribute to cancer prevention by providing community-based interventions, including one-on-one education and screening programs for breast, cervical, and colorectal cancers [30,31]. MHUs are also gaining traction relative to underserved populations or citizens in developing countries, where they help to overcome barriers and disparities such as transportation challenges, long waiting times, and limited health infrastructure [32,33,34,35,36,37,38]. In India, MHUs were introduced as early as 1951 to serve tribal areas, typically consisting of a physician, pharmacist, auxiliary nurse midwife, and paramedical staff [39]. In Saudi Arabia, MHUs increased treatment success rates for tuberculosis treatments, reducing mortality and loss to follow-up [40]. Beyond clinical care, MHUs support rapid assessments, health promotion campaigns, and long-term relationships with marginalized populations [39,41,42,43]. However, their success depends on effective integration with health systems, proper legislation, community trust, and culturally competent staff [44,45,46].

Given their proximity to underserved communities and the trust they often build, MHUs are well positioned to engage in prevention and health promotion efforts, particularly for chronic disease risk reduction and early diagnosis, which also reduce unnecessary hospital admission rates, hospital stays, and overall healthcare costs [3,5,26,47,48]. MHU personnel can also contribute to ongoing community-based care to improve adherence and compliance. Their engagement in social welfare and psychosocial initiatives can foster health equity, community resilience, and solidarity [49,50,51,52]. Additionally, MHUs can integrate digital tools into daily practice, improving communication, care continuity, and system efficiency [53,54,55,56,57]. To meet these multifaceted roles, MHU personnel must possess a combination of knowledge, critical thinking, digital competencies, and interdisciplinary collaboration skills [9]. Previous studies have demonstrated that continuous professional development is essential to maintaining and enhancing the quality of care provided by healthcare professionals [58,59]. Thus, MHU personnel need to cultivate essential skills and competencies related to prevention and care provision; social welfare and digital skills; and the necessary knowledge, critical thinking, and interdisciplinary capacities needed in order to deliver comprehensive, equitable, and digitally supported care that is in line with the principles of PHC and PH [59,60,61].

In Greece, hospitals and non-governmental organizations (NGOs) have successfully deployed mobile units to deliver healthcare to hard-to-reach populations—including island residents, the Roma community, and refugees—demonstrating their effectiveness in reducing urban–rural healthcare disparities and costs [62,63,64,65]. These initiatives emphasize the need for tailored, context-specific strategies to ensure equitable and sustained access to care [66]. In this respect, the Hellenic National Public Health Organization (HNPHO, EODY) seeks to capitalize on the experience of MHU personnel who engaged in the COVID-19 response and related programs, establishing a structured, well-trained team to provide daily, cost-free healthcare services within the community (including hard-to-reach populations and underserved remote areas) [67,68]. This initiative forms a key component of the national public health strategy and has been formally institutionalized through recent legislation by the Ministry of Health, as published in the Government Gazette (Series B, No. 4983/03.09.2024) [69]. To support the evolving role of MHUs, the HNPHO (EODY) prioritized the enhancement of its personnel’s knowledge and skills related to PHC and PH. This created the necessity to cultivate knowledge and skills simultaneously across a diverse workforce with varying backgrounds and expertise in health-related disciplines on a nationwide basis. The aim of this study is to present the development and evaluation of the educational program “Enhancing Competencies of Mobile Healthcare Units Personnel” implemented by the Center of Training and Lifelong Learning (CTLL) of the University of Crete (UoC) according to the needs of MHU teams of HNPHO (EODY). This initiative responds to the growing need for advanced, flexible, and structured training programs tailored to the complex role of MHUs in PHC and PH. The program was designed to strengthen knowledge and skillsets in key areas such as prevention, primary care delivery, social support, and digital literacy. Grounded in the principles of interdisciplinarity and community engagement, it also aimed to develop essential skills for prioritizing health issues, delivering collaborative care, and promoting social welfare and public trust. The program seeks to empower MHU professionals toward becoming competent, community-focused practitioners who are capable of meeting evolving public health needs.

## 2. Materials and Methods

### 2.1. Center of Training and Lifelong Learning

The UOC’s CTLL “https://www.kedivim.uoc.gr/ (assessed on 10 May 2025)” designs, implements, and certifies interdisciplinary continuing education programs, ensuring alignment with national and European lifelong learning frameworks. It focuses on adult reskilling and upskilling, providing accessible, high-quality training in diverse formats, including face-to-face, online, and blended learning. The preparation and implementation of educational programs are based on the national and European institutional frameworks for lifelong learning, and are also aligned with the Sustainable Development Goals [70]. The educational program was submitted for evaluation in terms of its purpose, content, educational objectives, modules, and trainers. The program was implemented by the UoC’s CTLL, based on a formal agreement with the HNPHO (EODY). It was submitted for institutional evaluation in terms of its aims, structure, content, educational objectives, modules, and teaching personnel, and was subsequently approved for implementation by the CTLL’s Council (Decision No 27622/06-12-2024, Project No 12076). The program also received approval from the UoC’s Institutional Research Board of the Special Account for Research Funds (IRB-ELKE) (Approval No 61618/18.12.2024). Educational programs are exempt from the need for bioethics committee clearance. All UoC CTLL programs adhere to GDPR requirements [71].

### 2.2. Program Design and Implementation

A nationwide, fully online training program entitled “Enhancing Competencies of Mobile Healthcare Units Personnel” was developed for MHU personnel operating under HNPHO (EODY). The design of the educational program followed a competency-based approach, and it is grounded in adult learning theory and the principles of interprofessional education. The curriculum was structured around four core thematic areas identified as critical to the MHU scope of operations: (i) preventive health (e.g., health screenings, vaccination protocols, blood pressure, diabetes); (ii) care provision (e.g., biomarker monitoring, control of vital functions, chronic disease management, treatment adherence); (iii) social welfare and solidarity initiatives (e.g., organ and blood donation awareness, psychosocial support, violence prevention counseling); and (iv) digital skill enhancement (e.g., telemedicine systems, digital documentation). The content was developed by the HNPHO’s program committee, which included academics, the organization’s members, and collaborating healthcare professionals. It consisted of 250 h educational modules provided either through daily live sessions (165 h in total) or as on-demand videos (85 h in total). The program was implemented from 16 January to 24 February 2025. The successful completion of the program required 200 h of participation for healthcare professionals and 80 h for administrative staff via live or on-demand video sessions. Figure 1 represents the curriculum and its components as it was designed and implemented.

### 2.3. Background and Expertise of MHU Members and Educators Within the Program

The MHU personnel of HNPHO (EODY) consists of 548 healthcare professionals from a range of specialties, primarily within the fields of PH and PHC. These include nurses, nursing assistants, health inspectors, medical laboratory technicians, and related specialties such as biologists. Figure 2A illustrates the different areas of expertise, and Figure 2B shows the geographical distribution of MHU personnel according to specialty across Greece. All MHU members are certified health professionals holding university or technical degrees, with practical experience in the provision of PHC. In addition to healthcare personnel, the MHUs are supported by 325 trained administrative staff, including management personnel who direct administrative operations and vehicle drivers responsible for field transportation.

To address the training needs of MHU personnel, the program committee emphasized the importance of an interdisciplinary team of trainers, representing diverse specialties and professional experience, in ensuring a comprehensive and integrative approach in developing the educational modules. The training team consisted of 64 professionals from all over Greece, including academics, medical doctors, nurses, and other healthcare providers such as dentists and pharmacists. This approach ensured a holistic perspective for the educational modules. Figure 2C presents the distribution of the 64 trainers by specialization.

### 2.4. Evaluation of Knowledge and Satisfaction

To evaluate the program’s impact on knowledge acquisition, a standardized 30-item multiple-choice test was administered both before (pre-test) and after (post-test) training. Questions were drawn from all four thematic areas to ensure comprehensive coverage. The primary outcome was the mean change in scores, indicating improvement in topic understanding. A student’s paired *t*-test was used to analyze including confidence intervals and effect sizes (Cohen’s d), to evaluate any statistical differences within the results.

### 2.5. Data Analysis

Participant satisfaction and perceived program quality were evaluated based on criteria such as content relevance, delivery format, clarity, and overall satisfaction. The assessment was conducted using a Likert scale ranging from “Poor” to “Excellent,” covering key aspects including program structure and organization, educational content, module quality and completeness, applied teaching methods, live sessions, on-demand video content, and trainer effectiveness. This assessment is part of the UoC’s Quality Evaluation Unit and is implemented across all CTLL programs to ensure consistent quality standards. The standardized questionnaire, which is same for all UoC’s CTLL programs, includes 11 open-ended questions to capture more detailed participant feedback, in addition to the Likert-style items. In parallel, trainers completed a separate evaluation form in order to provide structured feedback on program organization, content, participant engagement, and procedural aspects using a numerical Likert-type scale (1 = not at all satisfied to 10 = highly satisfied) along with an open-ended free text question inviting general comments on the training experience.

## 3. Results

### 3.1. Program Content and Learning Objectives in Accordance with CTLL Requirements for Adult Education

The educational program was approved for implementation by the CTLL Council (Decision No 27622/06-12-2024, Project No 12076). It was categorized under the CTLL section of Medicine and Health Sciences and aligned with multiple sustainable development goals such as good health and wellbeing, quality education, gender equality, sustainable cities and communities, peace, justice and strong institutions, and partnerships for the goals. As a CTLL program, its educational objectives encompassed cognitive, psychomotor, and behavioral dimensions. In terms of cognitive skills, the program aimed to develop participants’ ability to analyze and prioritize data through observation and prediction, address both existing and emerging health needs, ensure access to care with a focus on prevention and early intervention, and integrate services addressing communicable, non-communicable, and mental health risks. With respect to psychomotor skills, the program emphasized facilitating collaboration in prevention and health promotion, applying tools for monitoring and evaluation, and effectively using digital systems to support care delivery. Finally, in the behavioral and attitudinal domain, the program focused on promoting health literacy within communities and professional teams, fostering a sense of contribution and responsibility to public health and primary, supporting colleagues through knowledge-sharing, and building trust while upholding social justice in the delivery of services.

### 3.2. Implementation of the “Enhancing Competencies of Mobile Healthcare Units Personnel” Program

The educational content, covering 250 h, was distributed across four core themes, as shown in Figure 3A. Thematic areas such as prevention and health promotion and care provision received a larger share of the content, in accordance with their vital role in the activities of the MHU. The educational modules were provided through live sessions (167 h) or on-demand videos (83 h), and they were also categorized according to the core theme (Figure 3B,C). The live sessions were held daily from Monday to Friday, covering 6 h. of training modules. In terms of module distribution (Figure 3D), they were also related to the four core thematic areas. Under prevention and health promotion, modules included cancer prevention, smoking cessation, physical activity, cardiovascular health, infectious diseases, behavioral health, respiratory conditions, metabolic disorders, diabetes, and vaccination. With respect to the provision of care, the curriculum focused on primary care, cardiovascular health, diabetes management, respiratory conditions, metabolic diseases, neurodegenerative diseases, neuropsychiatric disorders, frailty, dental and sensory health, and biochemical testing. Regarding social welfare and solidarity initiatives, training covered behavioral health promotion, vaccination campaigns, blood donation, organ donation, and equity in health. Finally, in the area of digital skills, targeted modules emphasized the integration of digital competencies into primary care practices, disease management, vaccination record systems, and interdisciplinary collaboration initiatives. The alignment of the training modules with core public health competency domains, as outlined in the WHO-ASPHER Competency Framework for the Public Health Workforce in the European Region and the WHO Global Competency and Outcomes Framework for Universal Health Coverage, is presented in Appendix A [72,73].

A multidisciplinary approach was reflected in all thematic areas (Figure 3E), and academics contributed across all areas, with the strongest engagement observed in prevention and health promotion activities. Medical doctors engaged in modules for prevention, health promotion, and care provision, highlighting their clinical expertise in both proactive and reactive healthcare strategies. Healthcare providers focused on prevention and health promotion, with notable input in care provision and social welfare initiatives. Nurses contributed across all domains, particularly supporting care provision, social welfare activities, and digital health skills, emphasizing their versatile role in multidisciplinary healthcare delivery. HNPHO (EODY) members concentrated primarily on social welfare and solidarity initiatives, reinforcing their significant role in public health interventions and community outreach.

Considering the participation of MHU members during the daily live sessions, the average participation rate was approximately 90.6%, ranging from 92 to 100% during the initial days of the program. Although a slight decline in participation was observed during the mid-phase, it never fell below 78%, and participation levels returned to initial elevated levels toward the end of the program (Figure 3F). Healthcare professionals registered an average (±SD) total viewing time of 213.5 (±15.3) hours, distributed as 148.3 (±7.2) hours for live sessions and 65.2 (±4.03) hours for on-demand sessions. Administrative staff registered an average (±SD) total viewing time of 91.4 (±28.3) hours, with 50.3 (±23.1) hours for live sessions and 41.1 (±20.3) hours for on-demand sessions. Overall, 95.9% of MHU members (96.7% of healthcare professionals and 95.2% of administrative staff) successfully met the minimum educational attendance requirements.

### 3.3. Knowledge Acquisition

The effectiveness of the training program in enhancing knowledge was evaluated using a standardized 30-item multiple-choice test administered before and after the program. The test was designed to reflect the educational content of the core thematic areas (Figure 4A). The results of the pre- and post-test stages indicated significant improvements in participants’ understanding, which were observed across all categories, suggesting that the program was effective in achieving its educational goals (Figure 4B). Overall, participants’ post-test scores improved from 52.8% to 69.8% before and after training, reflecting a substantial increase in knowledge and a mean increase of 17% of correct answers across all questions after the intervention. This difference was statistically significant, with a 95% confidence interval (CI) of [12.9%, 21.0%], indicating a robust and positive shift in knowledge levels across the trainees. The standard deviation of the differences in percentage points (i.e., intra-item variability) was 10.90, suggesting moderate heterogeneity in the extent of improvement across individual questions. Specifically, scores in “Prevention–Health Promotion” increased from 47% to 65%, “Care Provision” increased from 54% to 68%, “Social Welfare and Solidarity Initiatives” increased from 68% to 83% and “Digital Skills” increased from 53% to 72%. The analysis of individual categories revealed statistically significant improvements (*p* < 0.05, 95% CI) for all categories, with the exception of “Social Welfare and Solidarity Initiatives”. Table 1 demonstrates the summary of learning gains by category, based on the percentage of correct answers before and after the educational intervention along with CI and Cohen’s d. Overall, the intervention resulted in an increase in correct responses across most items. Improvements were particularly prominent in questions related to “Prevention-Health Promotion and Digital Health”, suggesting enhanced comprehension in these domains. The full table is available in the Appendix A. These findings suggest that the program effectively supported the participants’ learning objectives and their comprehension of core PH and PHC topics relevant to the program’s core thematic areas.

### 3.4. Participant Satisfaction and Trainers’ Feedback

The trainees’ evaluation of the national training program, based on a Likert-scale survey from CTLL, reveals its broadly positive reception across all assessed dimensions (Figure 5). The majority of participants rated components such as “Structure and Organization”, “Program Content”, and “Educational Methods” as either “Very Good” or “Excellent,” with respective high ratings reaching 75.4%, 79.4%, and 80%. Similarly, “Live Educational Sessions” and “On-Demand Video Content” were well-received, with combined top-tier ratings of 82.3% and 76.8%, indicating strong appreciation for the program’s flexible delivery. “Trainers’ Effectiveness” received particularly high marks, with 80.3% of respondents rating this aspect as “Very Good” or “Excellent,” and the “Overall Program Evaluation” was also favorable, with 77.9% in the top two categories. Although a small percentage of trainees rated elements as “Poor” or “Very Poor” (ranging from about 8% to 11% across domains), the results suggest consistently high satisfaction levels. Notably, “CTLL Administrative Support” and “Management and Trainee Engagement” were also positively rated, with “Very Good” or “Excellent” responses reaching 78.4% and 72.3%, respectively. These findings underscore the program’s success in both content delivery and organizational execution, supporting its role as a model for scalable, multidisciplinary continuing education in primary care.

Trainers provided structured feedback on four key areas—organization, educational content, participant engagement, and procedural requirements—and overall, they expressed high satisfaction with the training program. On a 10-point Likert scale, the organization received a highest average rating of 9.4, with academics rating it as an outstanding 9.9, reflecting strong program structure and coordination. Educational content was also rated highly (8.8), underscoring its relevance and quality across professional groups. The engagement and participation of trainees received similarly positive scores, averaging 8.9, with academics and nurses particularly noting the interactive nature of the sessions. Procedural requirements—assessed on a scale where 1 denoted “difficult” and 10 denoted “easy”—averaged 3.2, demonstrating that the trainers found the administrative processes and online delivery manageable. In a free-text evaluation, trainers emphasized the professionalism of the organizing team, praised the training’s alignment with frontline public health needs, and highlighted the initiative as a successful model of academic–public health collaboration, recommending its continuation and expansion.

The open-ended feedback from trainees was highly diverse, containing numerous distinct suggestions for future training topics, as well as varied reflections on the program’s strengths. In contrast, trainer responses to the single open-ended question were consistently positive, emphasizing the program’s quality, relevance, and organization. Due to the large volume and heterogeneity of trainee feedback, no formal qualitative analysis was conducted at this stage. To account for this variability, the scope of the current work, and to avoid interpretation bias, open-text responses were not formally analyzed for inclusion in this study. Instead, a comprehensive thematic analysis is planned for future research.

## 4. Discussion

This study demonstrates that a structured online training program for MHU personnel can significantly enhance knowledge and professional competencies within the framework of PHC. Consistent with the global PHC goals of integrated, personalized care, the curriculum addressed both biomedical knowledge, such as chronic disease management, and the social dimensions of care [74,75]. Participants exhibited notable improvements in related knowledge and competencies, reflecting results from similar initiatives of continuing medical education (CME) programs [76]. These outcomes reinforce global PHC declarations, such as Alma-Ata and Astana, which advocate capacity-building and lifelong learning to deliver equitable, high-quality care [1,77,78]. Our approach supports previous evidence indicating that CME positively impacts healthcare professional performance and patient outcomes [79,80,81,82,83,84,85]. High levels related to participant satisfaction and the intention to apply new knowledge further suggest that CTLL-facilitated training can be an effective vehicle for advancing PHC objectives.

### 4.1. Key Findings

The educational program produced significant outcomes in terms of educational delivery, knowledge acquisition, and participant satisfaction. Its 250 hour structure, encompassing four thematic pillars, was tailored to the operational context of MHUs, with particular emphasis on prevention, health promotion, and care provision. This focus aligns with the population health mission of MHUs and the growing emphasis on preventive interventions. The delivery model, comprising 167 h of live content and 83 h of asynchronous learning, provided flexibility and interactivity, facilitating broad engagement across geographically and professionally diverse cohorts (Figure 2). Live sessions were especially valued for their interactivity and contextual relevancy, affirming the pedagogical value of synchronous learning in adult education. Attendance rates remained high (average 90.6%) despite a mid-program fluctuation, reflecting the training’s relevance and the commitment of both organizers and participants.

### 4.2. Discussion in Light of Literature

A major strength of the program was its multidisciplinary curriculum design, integrating contributions from academics, medical doctors, nurses, and other healthcare professionals (Figure 2 and Figure 3). This collaborative framework fostered cross-disciplinary learning and simulated the real-world teamwork required in mobile and community-based health settings. The integration of different perspectives enriched the educational experience, enhanced participants’ understanding of interprofessional roles, and cultivated a shared commitment to holistic care [86,87]. Pre- and post-training assessments suggest that the program effectively filled knowledge gaps and delivered essential components for MHU operations (Figure 4). Interpreted under the prism of Kirkpatrick’s Four-Level Model, our evaluation currently corresponds to Level 2 (learning outcomes), as evidenced by the statistically significant knowledge gains. These gains indicate that the training successfully improved the cognitive domain of professional competence, particularly in areas relevant to MHU’s role in PH and PHC in Greece. Future research efforts will aim to assess Level 3 (behavioral change) and Level 4 (impact on service delivery), particularly as the role of MHUs in Greece’s PHC continues to evolve [88,89]. This will involve exploring whether the knowledge and skills acquired are being applied effectively in the field and whether they contribute to measurable public health indicators.

The collaboration between the HNPHO and the UoC serves as a model for academic–public health partnership, highlighting the value of bridging theoretical knowledge with frontline practice. Such partnerships are increasingly recognized for their role in building resilient, equitable health systems [90,91,92]. Moreover, they align with previous studies emphasizing the importance of continuous, modular professional development in enhancing the capacity of primary care and community-based healthcare providers [93,94,95].

Although the program primarily assessed factual knowledge and clinical skills, it also acknowledged the need for advanced patient communication, empathy, and psychosocial assessment [96]. The biopsychosocial model holds that health and illness arise from an intricate blend of biological, psychological, and social factors; hence, healthcare providers should hold “soft” competencies that align with public health needs to address patients holistically [97]. The inclusion of psychosocial content (e.g., mental health, stressors, social determinants) reinforced the importance of addressing non-medical patient needs. This is consistent with advocacy for integrating clinical training with community-oriented perspectives, prioritizing advocacy, and empowerment alongside traditional care [98]. In light of this, the observed improvements in MHU staff members’ confidence in addressing both medical and psychosocial patient concerns are especially meaningful; this phenomenon suggests progress toward the WHO-defined PHC ideal of empowering individuals and communities through accessible, comprehensive care [78].

### 4.3. Positioning the Current Initiative of the Greek MHU Training Model Within the European Context

MHUs are increasingly seen as essential components of community-based healthcare delivery across Europe. Yet, despite their growing importance, training programs for MHU personnel remain fragmented and inconsistent in terms of scope, content, and delivery methods. In France, MHUs are regularly deployed for a wide range of services—from routine health screenings to vaccinations and emergency care—especially in rural areas and during public health crises [99]. Italy has implemented MHU programs since 2011 to enhance access to care for vulnerable populations such as migrants and individuals experiencing homelessness [100]. In Turkey, MHUs serve remote and high-need populations, including migrants, using facilities that range from basic clinics to fully equipped mobile hospitals. These services are delivered by a mix of government agencies, NGOs, and private hospital providers [101].

Across Europe, training for healthcare professionals working with migrants and ethnic minorities reflects a growing focus on cultural competence, intersectionality, and health equity, though program quality and structure vary significantly [102]. Countries such as Belgium, Germany, Moldova, and Greece have implemented initiatives to upskill health professionals in community-based, culturally responsive care for mental disorders [103,104]. Education and training programs offered by Schools of Public Health in France, Poland, Portugal, and the UK address key competencies in public health, including essential public health operations (EPHOs). However, there is considerable variation in the strength and consistency of these programs within and across countries [105]. Efforts to harmonize training at a regional level include the Erasmus + EPaCur project, which has developed a transnational paramedic curriculum to support mobility and standardization in Nordic countries [106]. In Hungary and Montenegro, pilot programs demonstrate how training in digital health and community engagement can effectively reach underserved groups [107]. Simulation-based mobile training units have also gained traction as a scalable solution. These mobile simulation units provide realistic, scenario-based learning directly within rural settings, enhancing retention and practical application of emergency and community care skills [108,109,110,111]. Additionally, digital and mobile technologies are increasingly used for peer learning and supervision. In this respect, advancements in AI offer promising tools for sustainable continuing education in healthcare, particularly through chatbots and related technologies that support digital literacy and behavioral change. While the effectiveness and appropriate use of these tools are still being explored, and their content must be critically assessed by both educators and learners, their potential should not be underestimated. Thoughtful integration of AI can complement traditional training and support the digital transformation of healthcare education [112,113].

Educational strategies to support the workforce of MHUs in Europe are vital for ensuring high-quality, equitable care in remote and underserved areas. However, European data remain sparse compared to global examples, underscoring the need for targeted investment in professional development infrastructures European healthcare systems [113]. A recent systematic review of community health services across Europe underscored the variation in organizational structures and the under-explored role of staffing and workforce dynamics, while highlighting potential benefits in reducing hospital re-admissions among vulnerable populations [114]. The integration of adult learning theory in these training programs acknowledges prior knowledge and aims to foster adaptive decision-making in complex field conditions [115]. The current educational program delivers a structured, multidisciplinary curriculum tailored specifically to the needs of HNPHO’s MHUs and the Greek healthcare ecosystem based on the principles and methods of adult learning and cultivation of microcredentials, integrating biomedical and psychosocial content in a flexible, blended online format. Interprofessional collaboration and soft skills training were central to the design and delivery of the modules. Leveraging the infrastructure of the UoC’s CTLL, the program presents a scalable national model. Despite certain limitations, it represents a promising example of adult education in community-based healthcare.

### 4.4. Strengths and Limitations

The use of the CTLLs to deliver this program underscores the role of institutionalized continuing professional development in strengthening healthcare delivery. By leveraging existing CTLL infrastructure within UoC, the initiative created a scalable model for staff education across the country. This approach is consistent with prior evidence that structured, competency-based training programs (often combining e-learning with practical components) result in significant gains in provider skill and patient care quality [116,117,118]. For example, a recent study showed that a blended certificate course in diabetes care produced statistically significant improvements in providers’ knowledge, clinical skills, and practice behaviors [76]. Similarly, sufficient evidence suggests that continuing education initiatives translate to better clinical performance and improved patient outcomes [119,120]. Notably, for chronic diseases such as diabetes, CME participation has even been linked to increased patient satisfaction and psychosocial wellbeing [121]. In addition to their effectiveness, educational interventions should also be economically sustainable, as suggested by previous studies on continuous professional development for healthcare professionals [122]. In this context, the current online program can be considered cost-effective, given the improvements in participant knowledge, the high level of participation, its nationwide implementation, and its timely completion. These results, along with high levels of participant satisfaction, highlight the program’s overall value and potential for broader application [123].

In our approach, ongoing CTLL-based education can help ensure that MHU staff members stay up to date with evidence-based guidelines and cultivate the interpersonal competencies that are necessary for effective community-level care. Despite the project’s results, some limitations must be acknowledged. First, the program’s online-only format, necessitated by logistical and geographical constraints, means that certain interactive or hands-on elements were limited. While e-learning offers flexibility and scalability, it may not fully replicate the richness of in-person workshops or clinical simulations [110,111]. Participants’ increase in knowledge was assessed immediately after the course but without later follow-up; in this case, we cannot confirm whether participants retained or translated what they learned into sustained practice. As the role of MHUs within the Greek healthcare ecosystem continues to evolve, future approaches could incorporate workplace-based assessments or health service data to track performance changes. Likewise, our evaluation focused on self-reported competence and test scores rather than objective clinical outcomes. We did not measure whether improved provider knowledge actually resulted in better patient health indicators (e.g., glycemic control, reduced complications) or more robust psychosocial support in practice. This could introduce selection or reporting bias in interpretation of the results. However, the high participation rates and the statistical analysis mitigate this concern. Finally, the study lacked a control group of MHU personnel with no training; thus, we cannot rule out alternative explanations for the observed improvements (such as participants’ initiative). The true impact of CME depends on how knowledge and skills are implemented in a practical context [120]. These caveats suggest caution in over-interpreting the immediate post-test gains without further evidence of long-term effects. While this study focused on qualitative and structured feedback data, the next phase of evaluation should incorporate interviews and observational data to triangulate outcomes. Moreover, the planned perspective studies should explore quasi-experimental or mixed-methods designs including control or comparator groups where feasible.

### 4.5. Impact and Perspectives

Overall, this study outlines the results of a national online training initiative aimed at enhancing the knowledge and skills of HNPHO (EODY) MHU personnel. Delivered through the infrastructure of the CTLL of UoC, the program highlights how structured, multidisciplinary educational efforts can be successfully implemented on a national scale to support PHC objectives and update the knowledge of healthcare professionals. The outcomes indicate meaningful knowledge acquisition among participants, underscoring the potential of such initiatives to strengthen workforce competencies in mobile, community-based settings. Future research should examine whether these educational improvements translate into real-world changes in care delivery. Evaluating how MHU staff apply new skills in routine practice and whether this leads to better health indicators—such as improved screening uptake, enhanced communication, or increased patient satisfaction—will be essential. Combining data with patient feedback through audits, surveys, or observational studies could offer a clearer understanding of the program’s broader impact, particularly in areas such as empathy, teamwork, and psychosocial care.

## 5. Conclusions

The educational program “Enhancing Competencies of Mobile Healthcare Units Personnel”, implemented by the CTLL of the UoC and EODY, represents a novel approach for continuous adult education and targeted intervention aimed at enhancing the knowledge and competencies of healthcare professionals on a nationwide scale. By placing public health at its core, the program strengthens the human resources of PHC through the systematic development of MHU personnel’s knowledge and skills, contributing to a more effective and community-responsive healthcare system. Moreover, it fosters a vital connection between frontline health services and the academic community, ensuring access to contemporary scientific knowledge and providing methodologies and tools for the holistic support of the population’s health. Finally, the program promotes innovation by encouraging cross-sectoral collaboration to meet the evolving needs of modern healthcare delivery.

## Figures and Tables

**Figure 1 healthcare-13-01706-f001:**
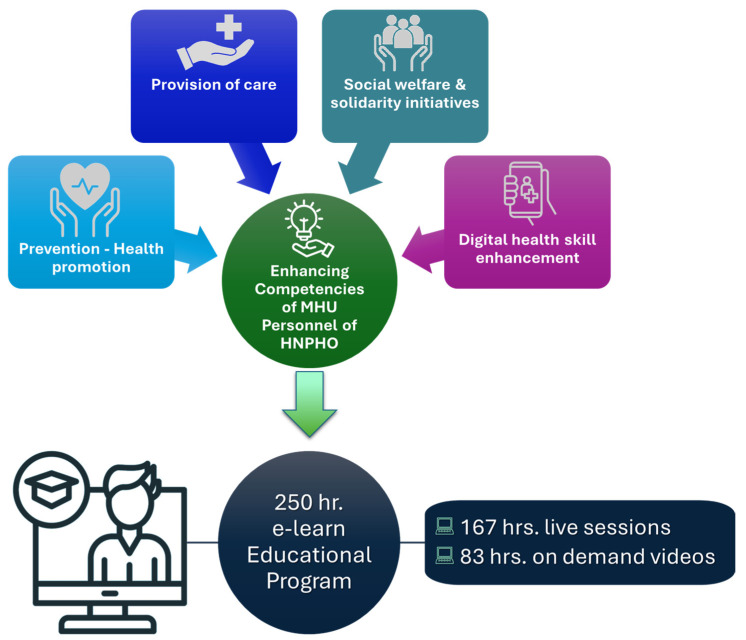
Design and implementation of the online educational program for the MHU members of HNPHO (EODY).

**Figure 2 healthcare-13-01706-f002:**
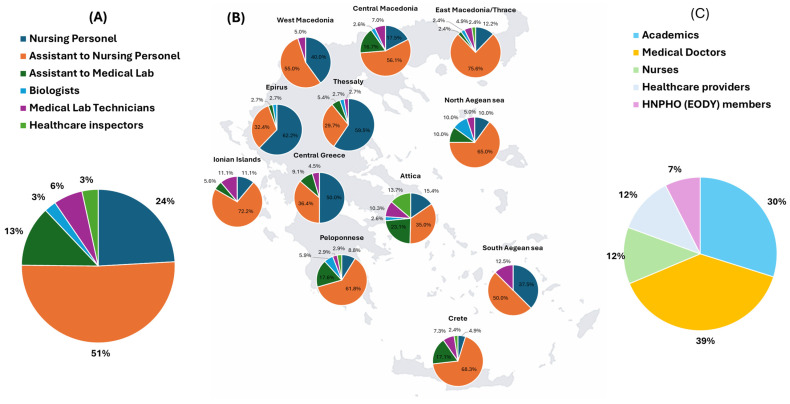
(**A**) Professional class of MHU members participating in the educational program. (**B**) Geographical distribution of the HNPHO (EODY) personnel across Greece. (**C**) Professional class distribution of trainers.

**Figure 3 healthcare-13-01706-f003:**
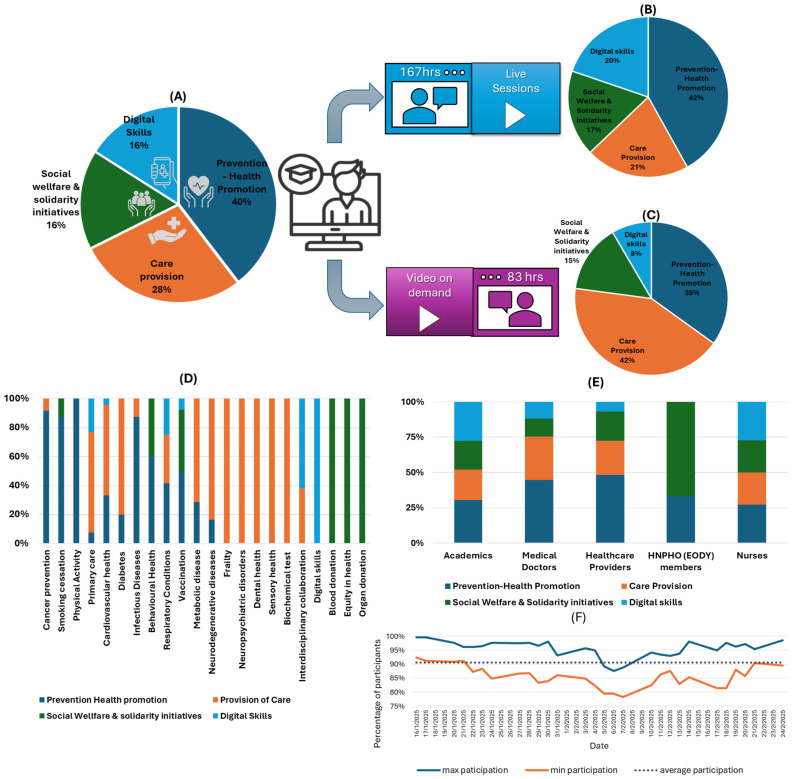
(**A**) Distribution of the 250 h training program per thematic area. (**B**,**C**) Distribution of training hours by delivery format (live vs. on-demand) and thematic area. (**D**) Classification of educational modules by general subject categories and their alignment with core thematic areas. (**E**) Trainer allocation per thematic area. (**F**) Daily participation in teaching modules by the MHU personnel.

**Figure 4 healthcare-13-01706-f004:**
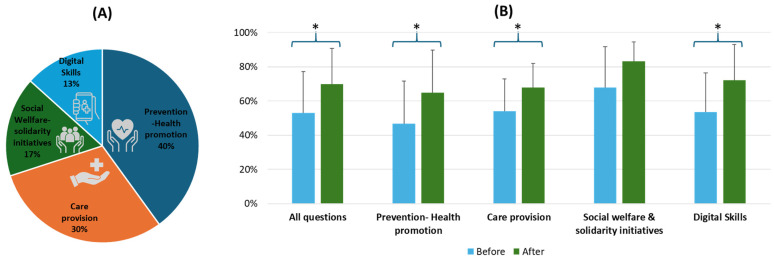
(**A**) Distribution of the 30-item multiple-choice test across the core thematic areas. (**B**) Average scores of participants before and after the program show an increase in correct answers. Statistically significant improvements are denoted by *.

**Figure 5 healthcare-13-01706-f005:**
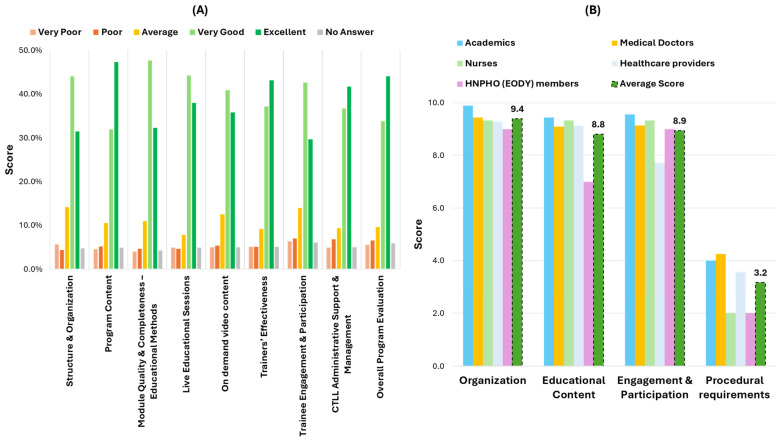
Program evaluation from (**A**) participants and (**B**) trainers.

**Table 1 healthcare-13-01706-t001:** Learning gain by category and comparison before and after the educational intervention.

Thematic Area	Mean % Before	Mean % After	Mean Improvement	95% CI	Cohen’s d	n
All Questions	52.8%	69.8%	+17.0%	[12.9%, 21.0%]	1.56	30
Health Promotion	46.6%	64.9%	+18.3%	[10.4%, 26.2%]	1.47	12
Care Provision	52.7%	68.0%	+15.3%	[8.3%, 22.3%]	1.69	9
Social Welfare and Solidarity Initiatives	67.8%	83.2%	+15.3%	[−2.4%, 33.0%]	1.08	5
Digital Skills	53.5%	72.0%	+18.6%	[5.8%, 31.3%]	2.32	4

## Data Availability

The data presented in this study are available upon request from the corresponding author (e.g., the data are not publicly available due to privacy or ethical restrictions).

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
