# Peer review of "Enhancing Competencies and Professional Upskilling of Mobile Healthcare Unit Personnel at the Hellenic National Public Health Organization"

_healthcare, 2025, doi:10.3390/healthcare13141706_

Round 1

Reviewer 1 Report

Comments and Suggestions for Authors

This article presents the development and evaluation of an online, nationwide, competency-based training program offered in 2025 to over 800 professionals employed in Mobile Healthcare Units (MHUs) in Greece. The training was structured in four thematic modules, prevention, healthcare provision, social care, and digital competences, and was evaluated via pre- and post-test scores and qualitative trainee and trainer feedback. The manuscript's scientific contribution is somewhat limited even though it is readable and based on a large population of participants, as it seems to not have a comparative or theoretical basis, and it reports findings descriptively without placing them in the context of current literature on adult learning, health workforce development, or e-learning assessment.

More importantly, it fails to deal with the deeper issue of how these findings are important beyond their institutional context, nor does it attempt to model or interpret critically the success of the intervention beyond basic before–after measures.

To improve it considerably, the following are some important suggestions:

  • As said, the manuscript does not have an apparent conceptual framework for training evaluation: without such a framework, the aim and outcome of the training are left to be taken at face value by the reader. The authors could situate their study in widely accepted training evaluation models, e.g., Kirkpatrick's Four-Level Model, other logic models, or continuing professional development in health services models. This would enable the authors to distinguish between knowledge improvement (Level 2), behavioral change (Level 3), and any impact on service delivery (Level 4), rather than merely focusing on score improvement without interpretive value.
  • The authors may utilize literature spanning the design, delivery, and effectiveness of digital education for frontline workers, specifically in public or mobile settings. For example, Cosma et al. (2025) offer a scoping review of chatbot-based platforms for enhancing vaccine literacy and uptake, with insightful information on digital tools as behavioral interventions, which can be utilized to guide the section on the digital literacy module in the manuscript. That study highlights the influence of perceived usefulness, users' trust, and interactivity of messages on uptake and literacy—factors that may also influence the digital component of this training.
  • The quantitative evaluation is based exclusively on pre-post testing with no external validation, behavioral follow-up, or assessment of long-term impact. There is no attempt to measure application of skills, skill retention, or impact on the quality of service. This limitation must be explicitly noted at least, and research trajectories outlined, e.g., mixed-methods studies or in-service behavioral assessments, to explore the actual influence of the program on practice.
  • The manuscript mentions participant and trainer feedback but presents no formal analysis of these qualitative inputs. Were these open-ended survey questions? Was there a formal coding process? Were dissident views aired? If qualitative data were sought, they must be formally analyzed and presented; otherwise, their mention is of little utility and can tokenize feedback.
  • Although the manuscript touches on MHUs in Greece to some extent, it does not explain how this model fits into other national contexts (e.g., Italy, France, or Turkey), nor does it discuss any of the literature on mobile health systems, equity in service delivery, or workforce mobilization issues. A clearer explanation of how this training model fits into (or diverges from) European patterns of MHU staffing and support would add significantly to the discussion.
  • The manuscript mentions "competency-based" training but does not explain what competencies were targeted, how they were measured, or why these were selected. Without operational definitions, the term is a hollow placeholder. The authors are encouraged to look to competency frameworks such as those developed by WHO or national public health agencies in order to base their modules on widely accepted terminology.

  • Cosma C, Radi A, Cattano R, Zanobini P, Bonaccorsi G, Lorini C, Del Riccio M. (2025). Exploring Chatbot Contributions to Improving Vaccine Literacy and Uptake: A Scoping Review of the Literature. Vaccine, 44, 126559. https://doi.org/10.1016/j.vaccine.2024.126559
  • Smidt A, Balandin S, Sigafoos J, Reed VA. The Kirkpatrick model: A useful tool for evaluating training outcomes. J Intellect Dev Disabil. 2009;34(3):266-274. doi:10.1080/13668250

Author Response

We thank the reviewer for their comments. Please find attached our responses 

Reviewer 2 Report

Comments and Suggestions for Authors
  • The paper contains some typographical errors such as repeated words or phrases (e.g., "MHUs had beenwere pivotal") 
  • Some issues with punctuation and formatting inconsistencies, including extra commas or irregular symbol use
  • limitations of Online-Only Format, absence of control group make difficult to know if the observed improvements were solely due to the training program or by other factors.
  • The improvement in effectiveness should take confounding factors into account.

Additional comments:

  • The study assessed knowledge gains immediately after the training program using pre- and post-tests, but it did not include any longer-term follow-up to determine the sustainability of improvements over time
  • Since the evaluation of programme was based on voluntary participation and self-reported measures, there could be a bias where only the most motivated individuals participated or provided positive feedback
  • To examine the program's full effectiveness, this paper would have benefited from a mixed-methods approach that incorporated long-term, practical outcome measures along with a proper control group availability. 

Author Response

We would like to thank the reviewer for the constructive comments that we received. Please find attached our responses to the points raised. 

Reviewer 3 Report

Comments and Suggestions for Authors

The manuscript aligns well with the journal’s scope and addresses a relevant and timely issue in public health. It presents a valuable case of skill development for mobile healthcare workers. However, several aspects of the study need refinement before the paper can be considered for publication. Although the initial findings are encouraging, there are key limitations. The absence of a control group, the lack of long-term follow-up, and the unclear description of the training design and assessment validation raise concerns about the strength and reliability of the results.

Recommendations for Improvement:

  • Clarify the rationale behind the study and explain how it addresses shortcomings in previous training programs.

  • Provide details on how the training was structured and whether the knowledge test used was validated or reviewed for quality.

  • Expand the evaluation beyond knowledge, and consider including other important competencies such as communication, teamwork, and digital literacy.

  • Incorporate qualitative feedback—such as participant interviews or open-ended responses—to offer deeper insight into how the training was experienced and perceived.

  • Report more comprehensive statistical details, including confidence intervals and effect sizes, to better support the significance of the results.

Comments on the Quality of English Language

The manuscript contains frequent minor language and formatting issues. Careful proofreading and editing are recommended to improve clarity and professionalism.

Author Response

We would like to thank the reviewer for the constructive comments and the positive evaluation of our work. Please find attached our responses to the comments raised. 

Round 2

Reviewer 1 Report

Comments and Suggestions for Authors

Thank you for the revised manuscript. The structure, clarity, and completeness have improved significantly, and the inclusion of full data and detailed analysis strengthens the overall presentation; the figures and tables are now well integrated, and the methodological description is clearer. That said, some in-text references still read awkwardly and disrupt the flow, I suggest revising for consistency and readability. With minor refinements to the language and citation style, the manuscript will be in strong shape for publication.

Author Response

Thank you for the revised manuscript. The structure, clarity, and completeness have improved significantly, and the inclusion of full data and detailed analysis strengthens the overall presentation; the figures and tables are now well integrated, and the methodological description is clearer. That said, some in-text references still read awkwardly and disrupt the flow, I suggest revising for consistency and readability. With minor refinements to the language and citation style, the manuscript will be in strong shape for publication.

Answer: We sincerely thank the reviewer for the constructive and positive feedback regarding the revised manuscript. We appreciate the recognition of the improvements in structure, clarity, and completeness, as well as the integration of data and methodological detail. In response to the minor comment, we further reviewed the manuscript and made the necessary edits to improve consistency, citation style, and overall readability.

Reviewer 3 Report

Comments and Suggestions for Authors

Thank you. The majority of the previously raised concerns have been addressed in a satisfactory manner. 

Author Response

Thank you. The majority of the previously raised concerns have been addressed in a satisfactory manner. 

We thank the reviewer for acknowledging the revisions and are pleased to hear that the majority of the previously raised concerns have been addressed satisfactorily